**Title Page**
Effects of land use/land cover and climate changes on surface runoff in a
semi-humid and semi-arid transition zone in Northwest China

4        Jing Yin [1], Fan He [2], YuJiu Xiong [3, 4, *], GuoYu Qiu [5, *]

5        [1] Research Center for Sustainable Hydropower Development, China Institute of

Water Resources and Hydropower Research, Beijing 100038, China.

7        [2] State Key Laboratory of Simulation and Regulation of Water Cycle in River

Basin, China Institute of Water Resources and Hydropower Research, Beijing 100038,
China.

10       [3] Department of Water Resource and Environments, School of Geography and

Planning, Sun Yat-Sen University, Guangzhou 510275, Guangdong, China.

12       [4] Department of Land, Air and Water Resources, University of California at Davis.

13       [5] Shenzhen Engineering Laboratory for Water Desalinization with Renewable

Energy, School of Environment and Energy, Peking University, Shenzhen 518055,
Guangdong, China.

First author Email address: yinjing@iwhr.com
* Corresponding author: YuJiu Xiong, Email address: xiongyuj@mail.sysu.edu.cn.
Tel./Fax: +86 20 84114575.
Co-corresponding author: GuoYu Qiu, Email address: qiugy@pkusz.edu.cn. Tel./Fax:
+86 755 26033309.

**Abstract**

Water resources, which are considerably affected by land use/land cover (LULC) and climate changes, are a key limiting factor in highly vulnerable ecosystems in arid and semi-arid regions. The impacts of LULC and climate changes on water resources must be assessed in these areas. However, conflicting results regarding the effects of LULC and climate changes on runoff have been reported in relatively large basins, such as the Jinghe River Basin (JRB), which is a typical catchment ($> 45000 \, km^2$) located in a semi-humid and arid transition zone on the central Loess Plateau, Northwest China. In this study, we focused on quantifying both the combined and isolated impacts of LULC and climate changes on surface runoff. We hypothesized that under climatic warming and drying conditions, LULC changes, which are primarily caused by intensive human activities such as the Grain for Green Program, will considerably alter runoff in the JRB. The Soil and Water Assessment Tool (SWAT) was adopted to perform simulations. The simulated results indicated that although runoff increased very little between the 1970s and the 2000s due to the combined effects of LULC and climate changes, LULC and climate changes affected surface runoff differently in each decade, e.g., runoff increased with increased precipitation between the 1970s and the 1980s (precipitation contributed to 88% of the runoff increase). Thereafter, runoff decreased and was increasingly influenced by LULC changes, which contributed to 44% of the runoff changes between the 1980s and 1990s and 71% of the runoff changes between the 1990s and 2000s. Our findings revealed that large-scale LULC under the Grain for Green Program has had an important effect on the hydrological cycle since the late 1990s. Additionally, the

conflicting findings regarding the effects of LULC and climate changes on runoff in
relatively large basins are likely caused by uncertainties in hydrological simulations.
**Keywords:** SWAT; climate change; land use/land cover; streamflow; Jinghe River
Basin.

**1 Introduction**

Both climate and land use/land cover (LULC) changes are key factors that can modify flow regimes and water availability (Oki and Kanae, 2006; Piao et al., 2007; Sherwood and Fu, 2014; Wang et al., 2014a). Since the 20th century, climate variability is believed to have led to changes in global precipitation patterns (IPCC 2007), thereby changing the global water cycle and resulting in the temporal and spatial redistribution of water resources (Milly et al., 2005; Murray et al., 2012). LULC changes are primarily caused by human activities (Foley et al., 2005; Liu and Li, 2008) and affect the partitioning of water among various hydrological pathways, including interception, evapotranspiration, infiltration, and runoff (Sterling et al., 2012). The influences of climate and LULC changes on hydrological processes and water resources will likely continue to increase, especially in arid and semi-arid regions characterized as vulnerable (Fu, 2003; Vorosmarty et al., 2010).

The impacts of LULC and climate changes on runoff can generally be identified by using hydrological models (Praskievicz and Chang, 2009). These models provide valuable frameworks for investigating the changes among various hydrological pathways that are caused by climate and human activities (Leavesley, 1994; Jiang et al., 2007; Wang et al., 2010). Distributed hydrological models, which use input parameters that directly represent land surface characteristics, have been applied to assess the impacts of LULC and climate changes on runoff in water resource management areas (Yang et al., 2008; Yang et al., 2014; Chen et al., 2015). The Soil and Water Assessment Tool (SWAT), a robust, interdisciplinary, and distributed river basin model, is

commonly used to assess the effects of management practices and land disturbances on
water quantity and quality (Gassman et al., 2007). The hydrological responses to LULC
and climate changes are often investigated through scenario simulations using the
SWAT model.
Although substantial progress has been made in assessing the impacts of LULC
and climate changes on water resources (Krysanova and Arnold, 2008; Vigerstol and
Aukema, 2011; Krysanova and White, 2015), most studies have focused on individual
factors (i.e., either LULC or climate); thus, the combined effects of LULC and climate
changes are not well understood because their contributions are difficult to separate and
vary regionally (Fu et al., 2007; D'Agostino et al., 2010; Wang et al., 2014a). For
example, some studies have suggested that surface runoff is affected more by climate
change (increased precipitation) than by LULC changes (Guo et al., 2008; Fan and
Shibata, 2015), and other studies have found that urbanization contributes more to
increased runoff than precipitation (Olivera and Defee, 2007). According to Krysanova
and White (2015), less than 30 papers were published between 2005 and 2014 on topics
related to the combined effects of LULC and climate changes and the SWAT model,
whereas 210 and 109 papers presented studies of climate and LULC changes,
respectively. However, water resource management requires an in-depth understanding
of the isolated and integrated effects of LULC and climate changes on runoff (Chawla
and Mujumdar, 2015).
Notable evidence of drying trends exists in semi-arid and semi-humid regions (Ma
and Fu, 2006; Li et al. 2007; Li et al. 2010; Li et al. 2011). These regions have
experienced serious water shortages in addition to intensive human activity and climate
change (Wang and Cheng, 2000; Ma and Fu, 2003). In this case, the effects of LULC
and climate changes on runoff are considerably more sensitive, and a dry climate can
result in serious environmental degradation and water crises (Ma et al., 2008; Jiang et
al., 2011; Leng et al. 2015). The Jinghe River Basin (JRB), which is located on the
central Loess Plateau, is a typical catchment located in a semi-humid and semi-arid
transition zone in Northwest China. The agricultural activities in this basin play an
important role in Northwest China (Zhao et al., 2014). However, the relative importance
of agriculture in the basin has caused ecological problems associated with social
development. For example, local water resources cannot maintain the rapid
socio-economic growth in the region (Wei et al., 2012), and the river system has
become unhealthy (Wu et al., 2014). Water and environmental management in the
region requires improved knowledge of the hydrological impacts of LULC and climate
changes. The effects of LULC and climate changes on the water cycle and water
resources must be assessed in these critical regions (Zhang et al., 2008; Li et al., 2009;
Qiu et al., 2011; Qiu et al., 2012; Peng et al., 2013).
Because the JRB transports the largest volume of sediment from the Loess Plateau
to the Yellow River, hydrological studies of the basin have primarily assessed the
impacts of soil and water conservation measures on surface runoff and sediment
transport (e.g., Feng et al., 2012; He et al., 2015; Peng et al., 2015a, 2015b; Wang et al.,
2016). Relatively few studies have been conducted regarding the effects of LULC and
climate changes on runoff. Studies of the Weihe River Basin (Zuo et al., 2014) and
Loess Plateau (Liang et al., 2015), which included the JRB as a sub-basin, have
identified the response of runoff to climate change and human activities by using a
climate elasticity model based on the Budyko framework. Zuo et al. (2014) found that
runoff in the JRB decreased by 17.79 mm between 1997 and 2009, with human
activities and climate change accounting for 51% and 39% of this decrease, respectively.
Liang et al. (2015) showed that streamflow decreased substantially from 1961 to 2009,
and the contribution of climate change (65%) to streamflow reduction was much larger
than that of ecological restoration measures (35%) in the JRB. Another study based on
the relationship between precipitation and runoff from 1966 to 1970 showed that runoff
mainly decreased due to precipitation before the 2000s and due to human activity
became dominant thereafter (with a contribution greater than 76%) (Zhang et al., 2011).
The different results reported by Zuo et al. (2014) and Liang et al. (2015) suggest that
assessing the impacts of LULC and climate changes on runoff in relatively large basins
(over 1000 km$^2$) is difficult (Chawla and Mujumdar, 2015; Peng et al., 2015b) due to
their complex effects on streamflow (Fu et al., 2007) and the variable boundary
conditions (Chen et al., 2011; Niraula et al., 2015).

Therefore, the objectives of this study were as follows: 1) to assess the surface

runoff variability influenced by LULC and climate changes in recent decades in the JRB
by using the SWAT model, which differs from the climate elasticity model based on the
Budyko framework; 2) to quantify the combined and isolated impacts of LULC change
and climate variability on surface runoff in the basin from 1971 to 2005 by using
scenario simulations after calibrating and validating the SWAT model at monthly and
yearly time scales; 3) to discuss how LULC and climate changes affect surface runoff;
and 4) to discuss the simulation uncertainty in the context of SWAT modelling due to
parameterizations and provide potential explanations for the conflicting results
regarding the effects of LULC and climate changes on runoff in relatively large basins.
**2 Methods and materials**
**2.1 Study area**
The JRB, which covers an area of approximately 45421 km$^2$, is located at 106°14′ –
108°42′ E and 34°46′ – 37°19′ N on the central Loess Plateau in Northwest China (Fig.
1). The main stream of the Jinghe River, with a length of 450 km, originates in the
Liupan Mountains in the Ningxia Autonomous Region and flows across Gansu and
Shanxi Provinces before draining into the Weihe River. The outlet gauging station,
Zhangjiashan, has a control area of approximately 43216 km$^2$. The study area is
characterized by hills and syncline valleys, with the Liupan Mountains to the west and
the Ziwu Mountains to the east. The elevation decreases from 2900 m to 360 m above
sea level. The climate varies from sub-humid to semi-arid, with mean annual
precipitation, temperature, and pan evaporation values of 390–560 mm, 8–13 °C, and
1000–1300 mm, respectively. Precipitation mainly occurs between July and September,
accounting for 50–70% of the total annual rainfall.
**2.2 Runoff change simulation**
Under the assumption that runoff is affected only by LULC and climate changes, the
effects of LULC and climate changes on surface runoff were evaluated using SWAT.
Before the simulations, the SWAT model was calibrated and validated as described
below.
**2.2.1 SWAT model and data collection**
SWAT, a semi-distributed hydrological model, was developed to assess the impacts of
land management and climate on water, nutrient, and pesticide transport at the basin
scale (Arnold et al., 1998; Neitsch et al., 2005). SWAT simulates hydrological processes
such as surface runoff at the daily time scale based on information regarding weather,
topography, soil properties, vegetation, and land management practices. In SWAT, the
study basin is divided into sub-basins, and each sub-basin is further subdivided into
hydrological response units (HRUs) with homogeneous characteristics (e.g., topography,
soil, and land use). Hydrological components are then calculated in the HRUs based on
the water balance equation.
In this study, SWAT is operated via an interface in ArcView GIS (Di Luzio et al.,
2002). Therefore, the required data are either raster or vector data sets, including a
digital elevation model (DEM), soil properties, vegetation, LULC, meteorological
observations, and discharge observations at Zhangjiashan gauging station.
(1) DEM
The Shuttle Radar Topography Mission (SRTM) 90-m DEM (Jarvis et al., 2008)
was used in this study.
(2) Soil data
Soil property information was obtained from the soil map of China at a scale of
1:1000000. The map was provided by the Chinese Natural Resources Database.
Huangmiantu, which covers 75.10% of the basin area, is the major soil type in the area

according to the Genetic Soil Classification of China. The other seven soil types are Heilutu (13.27%), Chongjitu (4.30%), Huihetu (3.23%), Hetu (2.41%), Hongniantu (1.10%), Cugutu (0.35%), and Shandicaodiantu (0.24%).

(3) Vegetation and LULC data

LULC data from four periods were retrieved from Landsat images by supervised classification, i.e., Multispectral Scanner (MSS) images (60 m resolution) from 1979, Thematic Mapper (TM) images (30 m resolution) from 1989, and Enhanced Thematic Mapper Plus (ETM+) images (30 m resolution) from 1999 and 2006. Each LULC dataset represents the land use patterns for one decade (e.g., LULC data obtained from 1979 represents the land use patterns in the 1970s). Land use was classified into seven categories: forest, dense grassland, sparse grassland, cropland, water, and barren areas. Then, the accuracy of the classification was verified, yielding a minimum Kappa coefficient of 0.73 (Xie et al., 2009).

(4) Meteorological data

Daily precipitation was collected from 16 rainfall stations (Fig. 1), whereas the daily minimum and maximum temperatures, wind speed, and relative humidity data required by the SWAT model were collected from 12 meteorological stations between 1970 and 2005. These data were interpolated to DEM grids using the SWAT model's built-in weather generator, which describes the weather conditions in the model simulations.

(5) Surface runoff

Daily runoff data measured at the Zhangjiashan gauging station between 1970 and

1990 were collected from the State Hydrological Statistical Yearbook. These data were
compared to the modelled surface flow during model calibration and validation.
**2.2.2 Model calibration and validation**
The SWAT model of the basin was first calibrated for the period of 1971 to 1997 and
was then validated for the period of 1981 to 1990. Based on published results (e.g., Li et
al., 2009) and our previous research results (Qiu et al., 2011), the simulation was the
most sensitive to the following six parameters: runoff curve number ($CN_2$), soil
evaporation compensation factor (ESCO), the available water capacity of the soil layer
(SOL_AWC), channel conductivity ($CH\_K_2$), the baseflow alpha factor (ALPHA_BF),
and the surface runoff coefficient (SURLAG). Therefore, these six parameters were
calibrated in the SWAT model (Table 1) (Qiu et al., 2011). Model performance was
assessed qualitatively using visual time series plots and quantitatively using the
coefficient of determination ($R^2$) and the Nash-Sutcliffe efficiency coefficient (*Ens*) (Eq.
(1)) (Moriasi et al., 2007).
$$Ens = 1 - \frac{\sum_{i=1}^{n}\left(Q_{obs} - Q_{sim}\right)^2}{\sum_{i=1}^{n}\left(Q_{obs} - Q_{obs\_m}\right)^2} \qquad (1)$$

where $Q_{obs}$ and $Q_{sim}$ are the observed and modelled runoff, respectively; $Q_{obs\_m}$ is the
mean value of observed runoff; and *n* is the number of data records. When *Ens*
approaches 1, the model simulates the measured data more accurately, whereas a
negative *Ens* indicates that the model performance is poor. In this study, a criterion
proposed by Moriasi et al. (2007), the Nash-Sutcliffe coefficient, was adopted to
evaluate the simulation (Table 2).
The SWAT model was calibrated and validated based on annual and monthly river
discharges measured at the outlet gauging station shown in Fig. 1.
**2.2.3 Simulation scenarios**
In this study, the effects of LULC and climate changes on surface runoff were evaluated
by comparing the SWAT outputs of ten scenarios. Each scenario represented one decade,
and each simulation required an LULC map and a meteorological data set (Table 3). If
the LULC map and the meteorological data were within the same decade (i.e., the 1970s,
1980s, 1990s, or 2000s), the simulation results represented "real runoff" or a "baseline"
affected by the combination of LULC and climate changes. Alternatively, varying one
driving factor while holding others constant simulated the effects of the variable factor
on runoff (Li et al., 2009). For example, to assess the response of streamflow to
combined LULC and climate changes in the 1970s and 1980s, the simulation of the
1970s (1970–1979) ($Q_{base,\ i}$), which is used as a reference period or baseline, should be
based on the current LULC (year 1979) and current climate (years 1970–1979). The
simulation of the 1980s (1980–1989) ($Q_{base,\ i+1}$) should be based on future LULC (year
1989) and future climate (years 1980–1989). The difference between the first and
second simulations represents the combined effects of LULC and climate changes on
streamflow. Regarding LULC changes, the third simulation ($Q_{sim,\ cL,\ i}$) was based on the
current climate (years 1970–1979) and the LULC in the next period, or the future LULC
(in this example, 1989). The difference between the first and third simulations is the
effect of the LULC change on streamflow. Similarly, the difference between the first
simulation and the fourth simulation ($Q_{sim,\ cc,\ i}$) based on the current LULC (year 1979)
and future climate (in this example, 1980–1989) represents the impact of climate change
on streamflow. The combined effects of LULC and climate changes on streamflow
($\Delta R_{comb}$%) and the isolated effects of LULC ($\Delta R_{iso, cL}$%) and climate ($\Delta R_{iso, cc}$%) can be
assessed using Eqs. (2) to (4).
$$\Delta R_{comb}\% = \left( \frac{Q_{base,i+1} - Q_{base,i}}{Q_{base,i}} \right) \times 100 \qquad (2)$$
$$\Delta R_{iso,cL}\% = \left( \frac{Q_{sim,cL,i} - Q_{base,i}}{Q_{base,i}} \right) \times 100 \qquad (3)$$
$$\Delta R_{iso,cc}\% = \left( \frac{Q_{sim,cc,i} - Q_{base,i}}{Q_{base,i}} \right) \times 100 \qquad (4)$$
**3 Results**
**3.1 Climate change**
Variations in precipitation, dryness index ($E_0/P$, defined as the ratio of annual potential
evapotranspiration calculated using the Penman–Monteith method to annual
precipitation), and air temperature were evaluated over four decades based on
meteorological data from 1970 to 2009 (Fig. 2). Precipitation decreased by 3.4% from
the 1970s to the 2000s. However, precipitation in the 1980s was slightly higher than that
in the 1970s. The decreasing trend in precipitation was substantial from the 1980s to the
1990s, reaching 4.1%. Thereafter, the decrease in precipitation was less than that from
1980 to 1999. During the entire period (from the 1970s to the 2000s), the temperature
increased by 13.6% (1.18 °C), including an abrupt increase of 0.7 °C from the 1980s to
the 1990s. Although the dryness index exhibited little change (increasing by 1.8%), a
large dryness index (>1.9) indicates that the climate became drier. These results indicate
that the climate in the JRB changed dramatically over the last four decades, as
characterized by decreased precipitation and increased temperature and dryness index
values. Both warming and drying trends are evident in the JRB. These results agree with
the results of other studies that reflect a broader phenomenon known as "climatic
warming and drying" in northern China (Ma and Fu, 2003; Huang et al., 2012).
**3.2 LULC change**
Figure 3 shows the variations in LULC distributions over the last four decades. The
dominant land-use types are sparse grassland (with a vegetation coverage of < 20%) and
cropland, which encompass a total of > 61% of the area over the four decades. However,
the percentage of sparse grassland was slightly higher than that of cropland, and the
margin varied from 2.96% to 9.80%. The remaining types include dense grassland (with
a vegetation coverage of $\geq$ 20%), forest, barren areas, urban and built-up areas, and
water, with mean ratios of 17.57%, 13.71%, 6.35%, 0.31%, and 0.29%, respectively.
The vegetation with low coverage that is predominant in the study basin corresponds
with the regional climate, and the relatively high percentage of cropland indicates the
importance of agriculture in this area.

The statistical results illustrated by the four LULC maps over the last four decades

indicate that vegetation (including grassland and forest) decreased by 11% between the
1970s and the 1990s and increased by 6% thereafter. The areas of cropland and urban
and built-up areas increased by 4.03% and 0.95%, respectively, over time. The area of
water fluctuated slightly, increasing by 0.09%. The area of barren land increased from
3.09% to 12.35% before the 1990s but then decreased to 3.02% in the 2000s. The
LULC changes potentially resulted from two major factors: social development and
population growth. These factors have increased since the 1980s, leading to the
expansion of urban and agricultural activities as well as unreasonable land utilization,
reclamation of vulnerable land, and vegetation removal. Therefore, the areas of urban
and barren land increased while the area of vegetation decreased. However, the
decreasing trend in vegetation changed due to a nationwide environmental conservation
programme initiated in 1999 by the Chinese government, the Grain for Green Program
(GGP) (Xu et al., 2004). The main goal of the GGP was to reduce soil erosion and
improve the eco-environmental status of western and northern China (Xu et al., 2004).
Noticeable evidence of ecological restoration was observed on the Loess Plateau after
the GGP was implemented (Chang et al., 2011; Sun et al., 2015). In addition to
preventing soil erosion, the GGP improved the soil physical and chemical properties
(Deng et al., 2014; Song et al., 2014) and facilitated vegetation restoration. The results
indicate that vegetation increased since the late 1990s, and these results agree with the
results of other studies (e.g., Liang et al., 2015; Wang et al., 2016).
**3.3 Performance of the SWAT model**
The SWAT model performed well in both the calibration and validation periods,
accurately simulating the outlet flows according to the model performance criteria ($R^2$
and *Ens*) after the six sensitive parameters were optimized. During the calibration
period (1971–1980), the time series plots of simulations and observations were similar
at both the annual (Fig. 4 (a)) and monthly scales (Fig. 5 (a)), although overestimation
was observed in the simulated streamflow. Point-by-point comparisons between the
simulations and observations further showed that most of the paired streamflow values
were distributed near the 1:1 line, with mean $R^2$ values of 0.90 (Fig. 4 (b)) and 0.84 (Fig.
5 (b)) at the annual and monthly scales, respectively (Qiu et al., 2011). In addition, the
results of a statistical analysis indicated that the mean *Ens* values were 0.76 and 0.72 at
the annual and monthly scales, respectively (Table 4). Similarly, although the SWAT
model did not perform as well during the validation period (1981−1990) relative to the
calibration period, the performance was still adequate, with *Ens* ($R^2$) values of 0.73
(0.83) and 0.69 (0.77) at the annual and monthly scales, respectively (Table 4, Figs. 6
and 7).

Although the *Ens* performance statistic associated with SWAT runoff modelling

can be larger than 0.8 in small or humid basins (e.g., Luo et al., 2008; Qiao et al., 2015;
Wu et al., 2016), *Ens* is typically less than 0.7 in relatively large river basins in arid to
semi-arid regions (e.g., Xu et al., 2011; Notter et al., 2013; Zhang et al., 2015; Liu et al.,
2016; Zhao et al., 2016). The *Ens* values in this study were generally good in the
calibration and validation periods and were comparable to those reported in other
studies in arid to semi-arid river basins. The results suggested that the SWAT model
performed well and was applicable to the study basin.
**3.4 Simulated surface runoff**
The annual runoff simulated by SWAT under different scenarios is shown in Table 3.
Generally, runoff increased minimally between the 1970s and the 2000s at a rate of 1.51
$m^3 s^{-1}$ (simulations S1 and S10) due to the combined effects of LULC and climate
changes (Fig. 8). However, runoff changed differently in different decades. For example,
runoff increased by 35.4% (29.75 $m^3 s^{-1}$) from the 1970s to the 1980s (simulations S1
and S4) but decreased thereafter. Notably, the simulated runoff in the 1990s was 12.59
$m^3 s^{-1}$ less than that in the 1980s (simulations S4 and S7), and runoff decreased by
15.5% (15.65 $m^3 s^{-1}$) from the 1990s to the 2000s (simulations S7 and S10) (Table 3).
**4 Discussion**
**4.1 Impacts of LULC and climate changes on surface runoff**
The hydrological effects were analysed using the simulated runoff data rather than the
observed data. The combined effects of LULC and climate changes on surface runoff
are presented in section 3.4. The simulated runoff increased between the 1970s and the
1980s, while precipitation increased from 521 mm to 527 mm during the same period.
Thereafter, runoff decreased as precipitation decreased. However, runoff decreased by
11.1% from the 1980s to the 1990s but decreased by 15.5% from the 1990s to the 2000s.
These results indicate that, although precipitation can considerably affect runoff
simulation, variations in runoff and precipitation were nonlinear due to the combined
effects.
The isolated impacts of LULC and climate changes on surface runoff can be
analysed by comparing two sets of simulations. The differences between S1 and S2 (as
well as between S4 and S5 and S7 and S8) reflect the impacts of climate change on
runoff. Accordingly, the differences between S1 and S3 (as well as between S4 and S6
and S7 and S9) reflect the impacts of climate change on runoff.
**4.1.1 Isolated impacts of LULC change**
During the first two decades, LULC changes increased runoff by 2.30 $m^3 s^{-1}$ and
accounted for 7.73% of the total change (29.75 $m^3 s^{-1}$). Thereafter, LULC change

decreased runoff by 6.83 $m^3 s^{-1}$, which accounted for 54.25% of the total change in runoff (12.59 $m^3 s^{-1}$) from the 1980s to the 1990s. The impacts of LULC changes on runoff increased during the last two decades because the contribution of LULC changes to runoff increased to 70.67% from the 1990s to the 2000s (Fig. 9).

The results in section 3.2 show that the LULC changed slightly from the 1970s to the 1980s. For example, the area of cropland marginally increased by 0.76%, and the vegetative area decreased by 3.19%. This small LULC change indicates that human activities minimally influenced runoff during the first two decades because the LULC changes only accounted for 7.73% of the increase in runoff. However, the LULC changed considerably with social development and population growth beginning in the 1980s. The vegetative area decreased by 7.81% from the 1980s to the 1990s, and the percentages of cropland, barren areas, and urban and built-up areas increased by 2.39%, 5.43%, and 0.11%, respectively. LULC changes associated with increased human activities accounted for 54.25% of the increase in surface runoff. Furthermore, the GGP, which was initiated in the late 1990s, mitigated the decreasing trend in vegetation. Although cropland and urban and built-up areas still expanded by 2.40% and 0.82%, respectively, from the 1990s to the 2000s, vegetation increased by 6.00%, and barren areas decreased by 9.33%. Therefore, LULC change exhibited a relatively large influence on the surface runoff change, contributing to 70.67% of the surface runoff in the last two decades.

In addition, the spatial distributions of different land use types influence the generation of runoff. As reported in our previous publication (Qiu et al., 2011), the soil

moisture content and evapotranspiration were modified by LULC changes (i.e., the GGP) after the GGP in the JRB, which led to changes in surface runoff. However, the modification was different. Fig. 10 shows that, after the GGP, the soil moisture content increased in the three selected sub-basins from the upstream to downstream regions, while the runoff and evapotranspiration decreased. When considering the upstream area as an example, barren land, with an initial percentage of 15.90%, and partial farmland, with an initial percentage of 6.56%, were converted to grassland due to the GGP, which improved water filtration and increased the soil moisture (Fig. 10 (a)). The simulation in Fig. 10 shows that the soil moisture content increased by 163.66%, 208.23%, and 262.66% in the sub-basins from the upstream to downstream, whereas the surface runoff (evapotranspiration) decreased by -37.53%, -38.55%, and -49.01% (-1.21%, -3.06%, and -25.90%), respectively. These results indicate that the impacts of LULC changes on flow regimes were larger in the downstream areas of the basin than in the upstream areas.

Although climate variables were held constant when simulating LULC changes, the isolated influences of LULC changes on runoff did not exclude the impacts of precipitation variations because the climate (including precipitation) varied in each decade (Table 3). Nonetheless, the above results indicate that LULC changes contributed considerably to decreased runoff, as reported in previous studies (e.g., Zhang et al., 2011; Zuo et al., 2014; Wang et al., 2014b; Wang et al., 2016). Additionally, the results suggest that vegetation restoration due to the GGP reduced surface runoff, which agrees with the results of other studies (e.g., Li et al., 2009; Nunes et al., 2011).

**4.1.2 Isolated impacts of climate change**

Unlike the contributions of LULC changes, the influences of climate change decreased in recent decades (Fig. 9). Climate change increased runoff by 26.07 $m^3 s^{-1}$ from the 1970s to the 1980s, accounting for approximately 87.63% of the increased total runoff during that period. Since the 1980s, surface runoff decreased, and the contributions of climate change to decreased runoff were 55.92% and 42.11% from the 1980s to the 1990s and from the 1990s to the 2000s, respectively. The influence of climate change on runoff agrees with climatic warming and drying trends. Decreasing precipitation will potentially lead to less runoff, whereas increasing temperatures will result in increased evaporation.

In summary, LULC and climate changes accounted for 7.73% and 87.63% of the total runoff increase (29.75 $m^3 s^{-1}$) in the 1970s and 1980s, respectively. The isolated influences of LULC and climate changes on runoff were nearly the same from 1980 to 1999 (54.25% and 55.92%, respectively) compared to the total decrease in runoff. In the last two decades, the percentage of the total runoff decrease that was caused by LULC changes (70.67%) was greater than that caused by climate change (42.11%).

Although uncertainties exist in the simulations (see section 4.2 for details), the above results indicate that the contribution of climate variability decreased over the last four decades, while the contribution of LULC change increased. Unlike the results reported by Liang et al. (2015), the findings in this study suggested that runoff fluctuations are influenced less by climate change and more by human activities. The results also indicate that the impacts of human activities on runoff have gradually

increased in the JRB, which agrees with the results of other studies (Zhang et al., 2011;
Zuo et al., 2014; Wang et al., 2016).
**4.2 Uncertainty in SWAT model simulations**
Uncertainty in model simulations, which is mainly caused by model structure (e.g.,
algorithm limitations) and model parameterizations, is a major challenge when
assessing the impacts of LULC and climate changes on runoff in relatively large basins.
In this study, the SWAT model performed well, with a Nash-Sutcliffe efficiency
coefficient and coefficient of determination of 0.76 and 0.90, respectively, for annual
runoff during the calibration period, as well as values of 0.73 and 0.83, respectively,
during the validation period. However, under the assumption that runoff is affected only
by LULC or climate changes, the simulated runoff associated with changes in only one
driving factor was slightly different than the simulated runoff obtained when
considering the combined effects of both factors due to the uncertainty in representing
LULC and climate change interactions in the SWAT model. For example, 28.37 $\mathrm{m^3\,s^{-1}}$,
which was the combined runoff rate in S2 and S3, was not equal to the "real or baseline
runoff" of 29.75 $\mathrm{m^3\,s^{-1}}$ in S4.
Qiao et al. (2015) reported that the SWAT model performed much better in small
watersheds (2–5 ha) than in a larger watershed (78 $\mathrm{km^2}$) because the meteorological
inputs (e.g., precipitation) do not represent the spatial variability in a given parameter
over larger basins because ground-based observations are limiting. To reduce the
uncertainty and improve the accuracy of the hydrological model and forecasting results
for relatively large basins, the uncertainty associated with model parameterization is
discussed below and potential solutions are proposed for future studies.
In this study, the basin area exceeded 45000 $km^2$. However, only 16 rainfall
stations were available, among which six stations were outside the study basin. The
station density was 0.35 stations per 1000 $km^2$. Xu et al. (2013) found that model
simulations are influenced by rainfall station densities below 0.4 per 1000 $km^2$. Under
such conditions, runoff simulations may contain uncertainties due to poor representation
of spatial precipitation variability, which is crucial in determining the runoff hydrograph
(Singh, 1997). Previous studies (e.g., Chu et al., 2011; Masih et al., 2011; Shope and
Maharjan, 2015) have suggested that the density of rainfall measurement stations has a
significant impact on SWAT simulations and that reducing the precipitation uncertainty
can improve the accuracy of simulated streamflows. Although the SWAT model
performed well in this study and the uncertainty in the simulations associated with
precipitation was similar to the uncertainties observed in other studies, peak flow
overestimation was observed in the simulated runoff (Figs. 4 to 7). To reduce
uncertainty, precipitation from stations should be processed (e.g., via interpolation)
before conducting runoff simulations, thereby improving the precision and spatial
representativeness, especially in relatively large basins without reliable and precise areal
rainfall data.
In addition, the coarse vegetation information provided by the LULC data in this
study can lead to uncertainty in the simulations because vegetation distinction is
required in SWAT modelling. Although the LULC data had a relatively high resolution
of 30 m, we can only provide a general vegetation categorization, such as forest, due to
the data limitations. Recent results (e.g., Pierini et al., 2014; Qiao et al., 2015) have
shown that detailed biophysical parameters of vegetation species can improve the
performance of distributed, physically based models such as SWAT and reduce model
uncertainty. In China, detailed and reliable data related to vegetation species are
uncommon. Reliable maps of vegetation species (as well as other geographic maps) at
high spatial resolutions (e.g., <1000 m) are an urgently needed to provide detailed and
heterogeneous information for accurate biophysical and hydrological parameterization.
**5 Conclusions**
In this study, the SWAT model was used to simulate the effects of LULC and climate
changes on surface runoff. The satisfactory performance of the SWAT model was
confirmed by the Nash-Sutcliffe coefficient and coefficient of determination values of
annual runoff of 0.76 and 0.90, respectively, during the calibration period and 0.73 and
0.83, respectively, during the validation period. Simulations showed that the combined
effects of LULC and climate changes increased surface runoff by 29.75 $m^3 s^{-1}$ during
the 1970s and the 1980s, whereas LULC and climate changes both decreased runoff by
28.24 $m^3 s^{-1}$ during the 1980s and the 2000s. Further analysis suggested that different
driving factors had different influences on surface runoff.
The isolated results indicated that the impacts of LULC changes on the
hydrological cycle were gradual, and that LULC changes altered runoff to a similar or
greater extent than climate change, accounting for 70.67% of the streamflow reduction
since the late 1990s. This result suggests that LULC plays an important role in the
transition zone between semi-humid and semi-arid regions. As an indicator that is
closely related to human activities, the LULC in the study area underwent considerable
changes, especially the vegetation cover rate, which decreased by 16% from the 1970s
to the 1990s and increased by 6% between the 1990s and the 2000s due to the Grain for
Green Program (GGP). In conclusion, the increased vegetation and land use changes
inevitably altered the hydrological cycle, and large-scale LULC changes under the GGP
considerably affected the hydrological cycle.
To reduce simulation uncertainty and improve the accuracy of hydrological
modelling and forecasting in relatively large basins, areal input parameters (e.g.,
precipitation and vegetation species information) should be generated with reliable
precision and high spatial resolution.
**Acknowledgements**
This study was supported by the National Natural Science Foundation of China
(Nos. 51309246 and 31300402), the China Scholarship Council (No. 201606380186),
and the National Basic Research Program of China (project No. 2006CB400505). We
thank the China Meteorological Administration for providing meteorological data. We
are grateful to the editors and reviewers for their insightful and constructive comments.

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

**Table 1.** Calibrated values of the six parameters in SWAT

| No. | Parameter name | Description | Range | Calibrated value |
|---|---|---|---|---|
| 1 | $CN_2$ | SCS runoff curve number for moisture condition II | -8–+8 | -8 |
| 2 | ESCO | Soil evaporation compensation factor | 0–1 | 0.1 |
| 3 | SQL_AWC | Available water capacity of the soil layer | 0–1 | 0.05 |
| 4 | $CH\_K_2$ | Channel conductivity | 0–150 | 0.35 |
| 5 | ALPHA_BF | Baseflow alpha factor | 0–1 | 0.01 |
| 6 | SURLAG | Surface runoff coefficient | 0–10 | 0.85 |



**Table 2.** SWAT performance of runoff simulations according to the Nash–Sutcliffe
coefficient (Moriasi et al., 2007).

| Simulation performance | Nash–Sutcliffe coefficient (*Ens*) |
|---|---|
| Very good | $0.75 < Ens \leq 1.00$ |
| Good | $0.65 < Ens \leq 0.75$ |
| Satisfactory | $0.50 < Ens \leq 0.65$ |
| Unsatisfactory | $Ens \leq 0.50$ |



**Table 3.** Simulated annual runoff by SWAT under different scenarios considering

both LULC and climate.

| | Scenarios | Climate | LULC | Simulation $(m^3 s^{-1})$ | Runoff change $(m^3 s^{-1})$ | Runoff change (%) |
|---|---|---|---|---|---|---|
| S1 | LULC and meteorological data from the 1970s | 1970s | 1970s | 84.10 | – | – |
| S2 | Changing LULC while holding climate constant | 1970s | 1980s | 86.40 | +2.30 | +7.73 |
| S3 | Changing climate while holding LULC constant | 1980s | 1970s | 110.17 | +26.07 | +87.63 |
| S4 | LULC and meteorological data from the 1980s | 1980s | 1980s | 113.85 | +29.75 | – |
| S5 | Changing LULC while holding climate constant | 1980s | 1990s | 107.02 | -6.83 | -54.25 |
| S6 | Changing climate while holding LULC constant | 1990s | 1980s | 108.61 | -7.04 | -55.92 |
| S7 | LULC and meteorological data from the 1990s | 1990s | 1990s | 101.26 | -12.59 | – |
| S8 | Changing LULC while holding climate constant | 1990s | 2000s | 90.20 | -11.06 | -70.67 |
| S9 | Changing climate while holding LULC constant | 2000s | 1990s | 94.67 | -6.59 | -42.11 |
| S10 | LULC and meteorological data from the 2000s | 2000s | 2000s | 85.61 | -15.65 | – |

**Table 4.** Nash-Sutcliffe coefficient (*Ens*) statistics in the SWAT calibration and
validation periods.

| Statistic | Calibration from 1971–1980 | | Validation from 1981–1990 | |
|:---:|:---:|:---:|:---:|:---:|
| | monthly | yearly | monthly | yearly |
| $N$ | 120 | 10 | 120 | 10 |
| Minimum | 0.58 | 0.53 | 0.54 | 0.58 |
| Maximum | 0.95 | 0.98 | 0.81 | 0.9 |
| Mean | 0.72 | 0.76 | 0.69 | 0.73 |



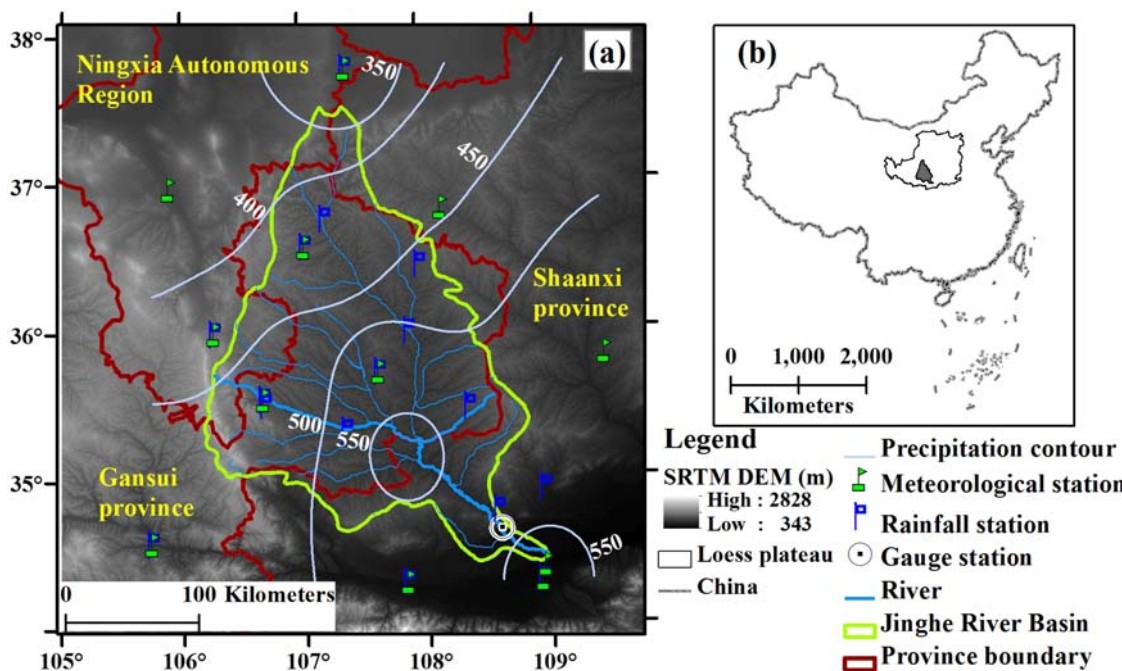


**Figure 1.** Geographic information regarding the study area: (a) Location and SRTM
DEM of the Jinghe River Basin and (b) schematic of the selected study area in China.
Precipitation (mm) is averaged and interpolated from meteorological data between 1970
and 2010.

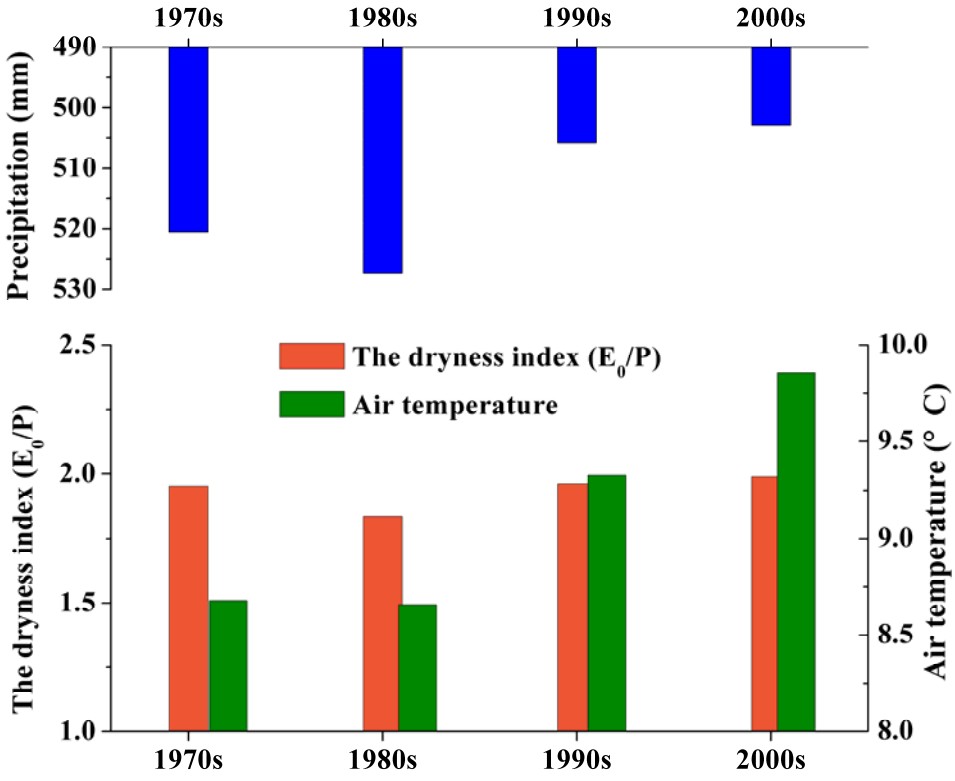

**Figure 2.** Variation in decadal mean precipitation (top), dryness index, and air temperature (bottom) in the Jinghe River Basin from the 1970s to the 2000s. The dryness index was defined as the ratio of annual potential evapotranspiration ($E_0$) to annual precipitation (P).

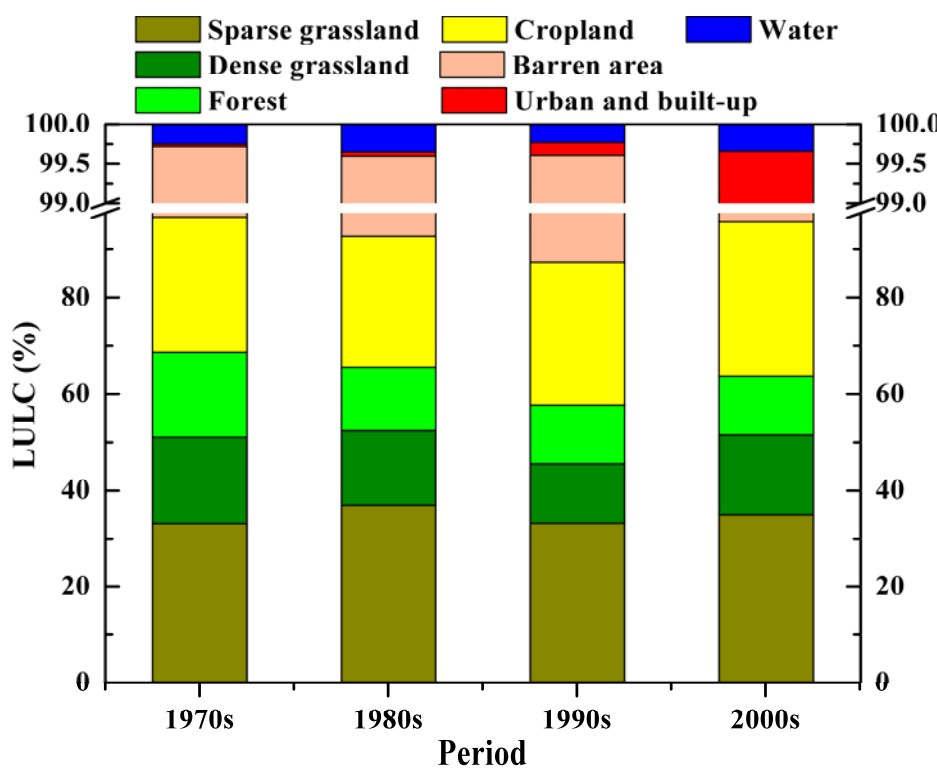


**Figure 3.** LULC composition and its change in the Jinghe River Basin from the 1970s
to the 2000s.

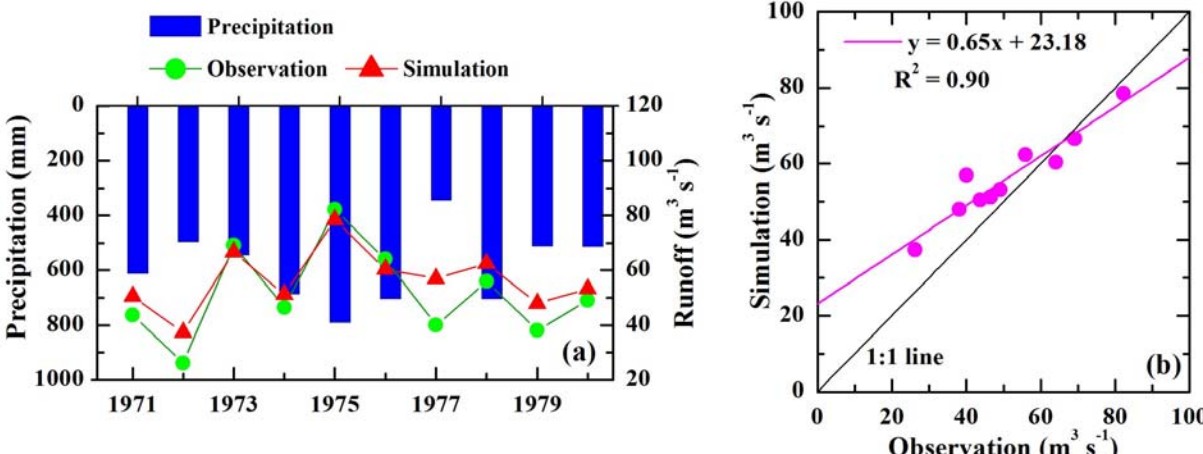


**Figure 4.** Comparison of observed and simulated runoff at the yearly scale in the

Jinghe River Basin during the calibration period from 1971 to 1980. Fig. 4(b) is redrawn

from Qiu et al. (2011).



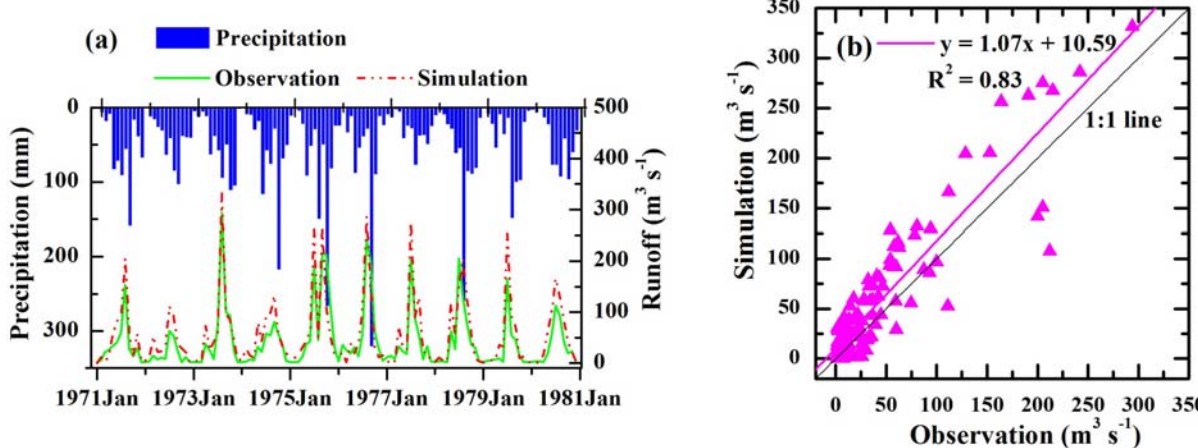


**Figure 5.** Comparison of observed and simulated runoff at the monthly scale in the

Jinghe River Basin during the calibration period from 1971 to 1980. Fig. 5(b) is redrawn

from Qiu et al. (2011).


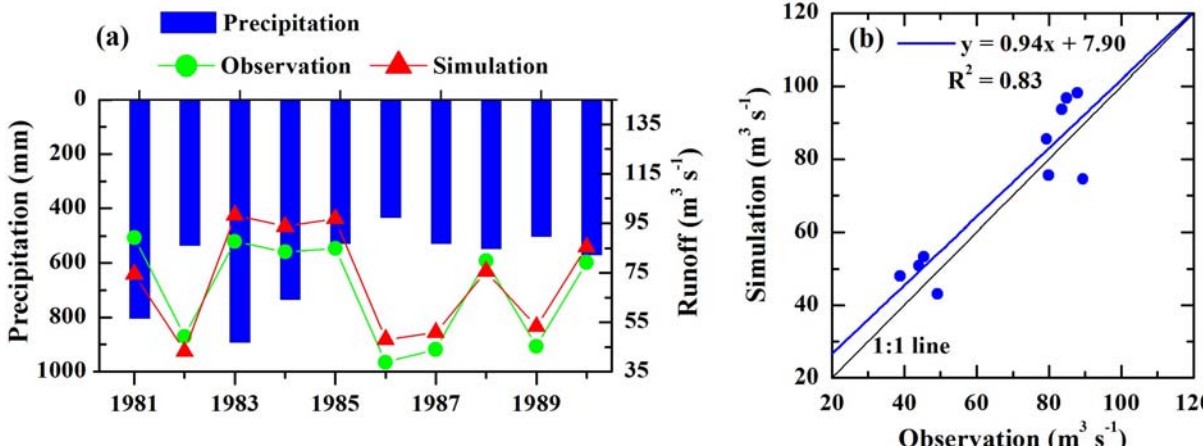

**Figure 6.** Comparison of observed and simulated runoff at the yearly scale in the Jinghe River Basin during the validation from 1981 to 1990. Fig. 6(b) is redrawn from Qiu et al. (2011).

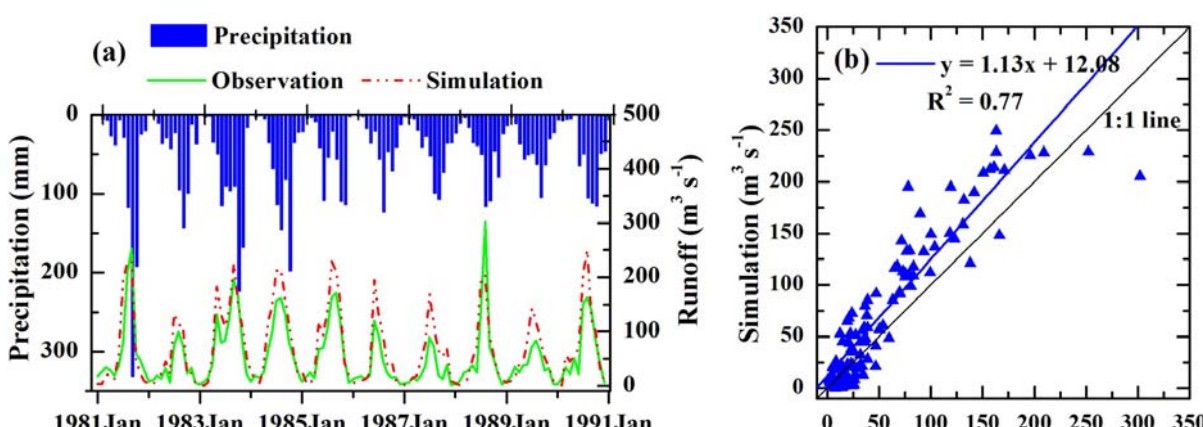

**Figure 7.** Comparison of observed and simulated runoff at the monthly scale in the Jinghe River Basin during the validation period from 1981 to 1990. Fig. 7(b) is redrawn from Qiu et al. (2011).

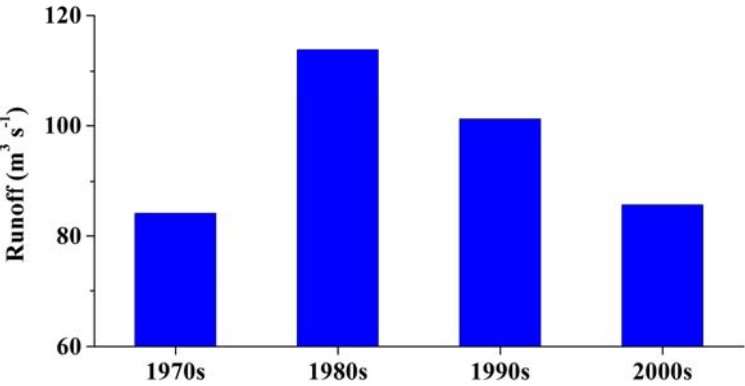


**Figure 8.** Variation in mean annual surface runoff at the decadal scale in the Jinghe

River Basin from the 1970s to the 2000s.


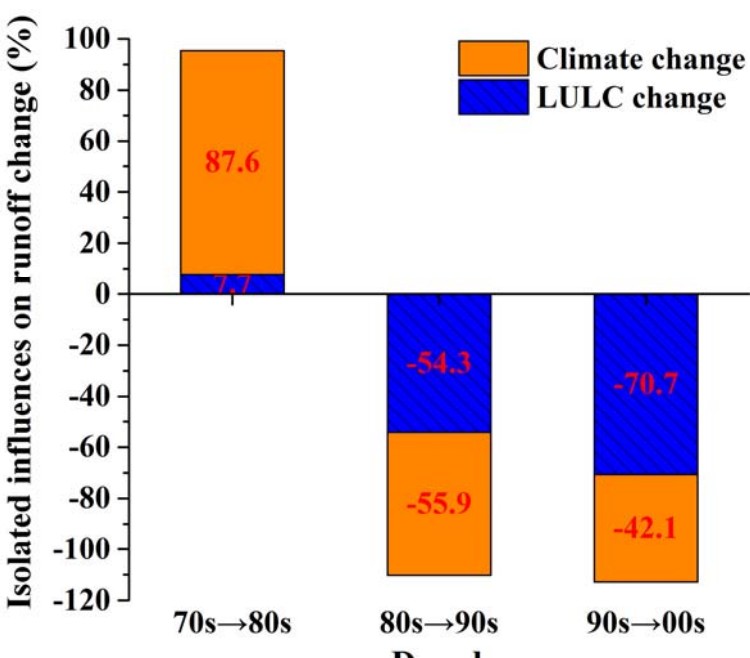


**Figure 9.** Isolated impacts of LULC and climate changes on surface runoff. Positive

values indicate that runoff increased due to these factors, whereas negative values

indicate that runoff decreased due to these factors. The summation of the isolated

influences is not equal to 100% due to simulation uncertainty (see section 4.2 for details).

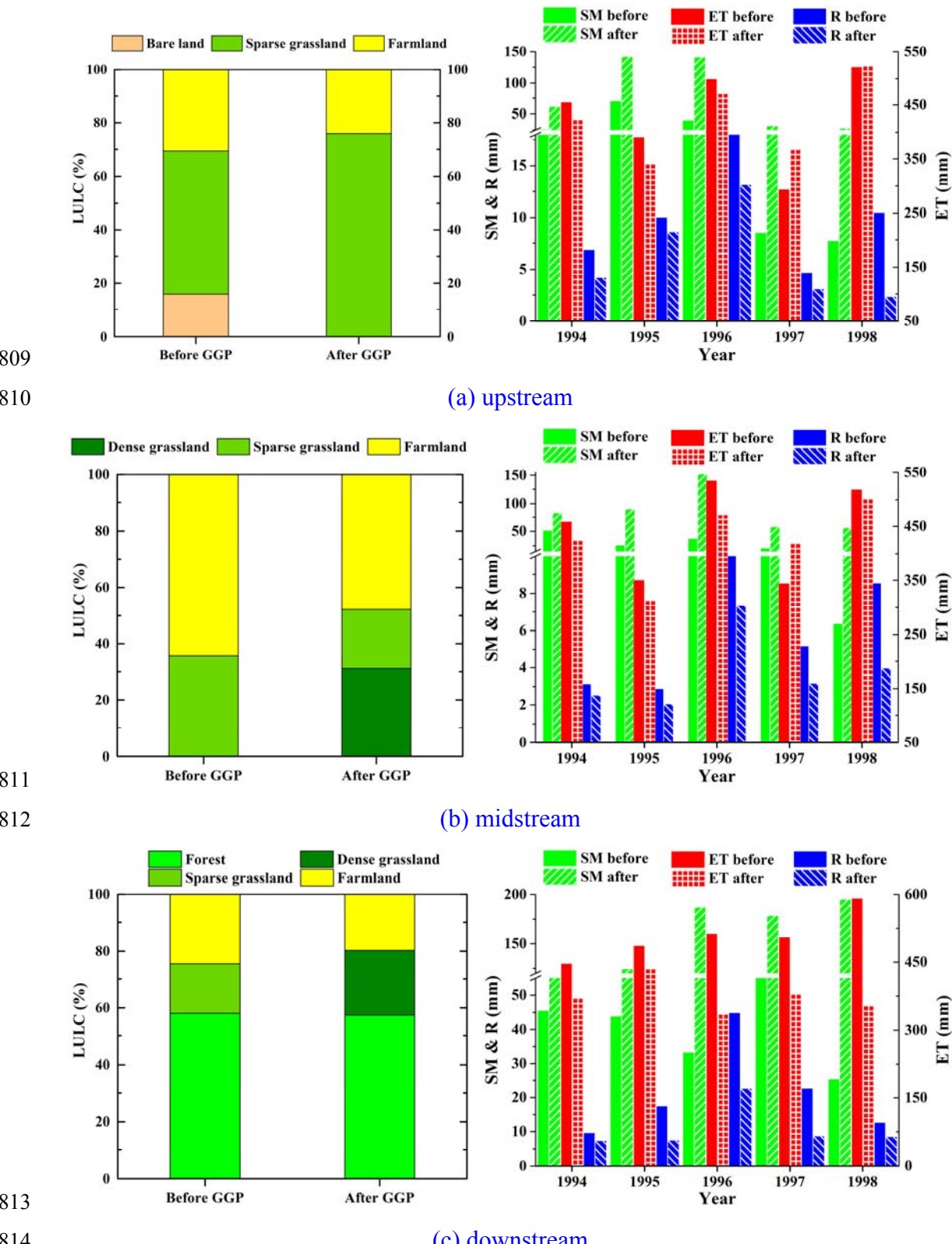


(a) upstream

(b) midstream

(c) downstream
**Figure 10.** Impact of LULC changes on surface runoff in selected sub-basins
distributed in the upstream, midstream, and downstream areas of the basin. The left
column shows the land use types and corresponding ratios, and the right column shows
the simulated changes of the soil moisture content (SM), evapotranspiration (ET), and
surface runoff (R) before and after the Grain for Green Program (GGP) scenarios while

holding climate constant.