# Peer review of "Title Page 1"

_Hydrology and Earth System Sciences, 2016_

## Referee Comment (RC1) · Anonymous Referee #1 · 16 Aug 2016

This manuscript investigates the relationships between land use and climate changes and corresponding hydrological responses in northwest China. The paper reveals some interesting findings. The manuscript can be considered for publication, if the following comments can be addressed properly. Major comments: 1. The effects of land use change and climate change should be discussed separately. 2. To explicitly assess the impact of land use changes on runoff generation, the impact of rainfall variation should first be excluded. Relevant discussion should be included on this topic. 3. I would encourage authors to rewrite the methodology description section. Give a clear message to the reader what you did and how you did. Some parts in the results analysis and discussion section (e.g. model calibration and validation) are more suitable to be in the methodology section. 4. The values in Table 2 are not acceptable, the authors should re-check the reference (Moriasi et al., 2007). 5. The influences of different land use types (such as area and spatial distribution) are also important to runoff generation. How to assess these effects? 6. It seems that the authors discussed the effects of the "Conversion of Cropland to Forest and Grassland Program" on the water budget of the Jinghe River basin (Qiu et al., 2011, Journal of Environmental Quality, 40, 1–11). What's the difference between the previous publication and this study?

Minor comments:

1. 3-28. The objectives. I would suggest the authors to add an objective to discuss how the LULC and climate changes affect surface runoff. 2. 4-6. Although finally draining into the Yellow River, the Jinghe River is a tributary of the Weihe River. 3. 5-10. How many runoff data was used and who performed the measurement? 4. 5-18. 'determination coefficients' should be 'determination coefficient'. 5. 8-31. I suggest the authors unify the number of decimal places. 6. 9-32. The citation style for two authors. The authors sometimes use 'and', but sometimes '&'. I suggest the authors unify the citation style. 7. The language need to be improved. There are many grammatical and spelling mistakes

---

## Author Comment (AC1) · 31 Aug 2016

Dear Reviewer(s):

Thank you for carefully reading the manuscript and providing constructive suggestions and comments. We appreciate your time and effort in considering the manuscript for publication.

All of the questions/comments have been carefully addressed in the revised manuscript. In this revision, the newly added content is blue, and the revised content is red.

The following are the point-by-point answers to each question/comment.

**Reviewer (s) Comments and our Responses**

**Reviewer: 1**

This manuscript investigates the relationships between land use and climate changes and corresponding hydrological responses in northwest China. The paper reveals some interesting findings. The manuscript can be considered for publication, if the following comments can be addressed properly.

**Major comments**

**1.** The effects of land use change and climate change should be discussed separately.

**Responses:**

Thank you for the valuable comments. We reorganized section 3 ('Results and Discussion') and divided section 3.4 ('Impacts of LULC and climate changes on surface runoff') into two sections, including section 3.4 ('Combined impacts of LULC and climate changes on surface runoff') and section 3.5 ('Isolated impacts of LULC and climate changes on surface runoff'). The new sections are as follows:

**3.4 Combined impacts of LULC and climate changes on surface runoff**

The annual runoff simulated by SWAT under different scenarios is shown in Table 3. The hydrological effects were analysed using the simulated runoff rather than the observed data.

Generally, runoff increased minimally between the 1970s and the 2000s, with a rate of 1.51 $m^3 s^{-1}$ (simulations S1 and S10) due to the combined effects of LULC and climate changes (Fig. 8). However, runoff changed differently in different decades. Runoff increased by 35.4% (29.75 $m^3 s^{-1}$) from the 1970s to the 1980s (simulations S1 and S4), but a decrease was observed thereafter, e.g., the simulated runoff in the 1990s was 12.59 $m^3 s^{-1}$ less than that in the 1980s (simulations S4 and S7), and runoff decreased by 15.5% (15.65 $m^3 s^{-1}$) from the 1990s to the 2000s (simulations S7 and S10) (Table 3).

The simulated runoff increased between the 1970s and the 1980s, while precipitation increased from 521 mm to 527 mm in the same period. Thereafter, runoff decreased as precipitation decreased. However, runoff decreased by 11.1% from the 1980s to the 1990s but decreased by 15.5% from the 1990s to the 2000s. The results indicate that although precipitation can considerably affect runoff simulation, variations in runoff and precipitation were nonlinear due to the combined effects.

[Figure]

**Figure 8.** Variation in mean annual surface runoff at the decadal scale in the Jinghe River Basin from the 1970s to the 2000s.

**3.5 Isolated impacts of LULC and climate changes on surface runoff**

The influences of LULC changes between adjacent decades can be analysed by comparing two sets of simulations (e.g., S1 and S2, S4 and S5, and S7 and S8). Accordingly, the differences between S1 and S3 (as well as between S4 and S6 and S7 and S9) reflected the impacts of climate change on runoff.

**3.5.1 Impacts of LULC change**

In the first two decades, LULC changes increased runoff by 2.30 $m^3 s^{-1}$, accounting for 7.73% of the total change (29.75 $m^3 s^{-1}$). Thereafter, LULC change decreased runoff by 6.83 $m^3 s^{-1}$, accounting for 54.25% of the total change (12.59 $m^3 s^{-1}$) from the 1980s to the 1990s. The impact of LULC change on runoff increased in the last two decades, because the contribution of LULC change to runoff increased to 70.67% from the 1990s to the 2000s (Fig. 9).

The results in section 3.2 showed that LULC changed slightly from the 1970s to the 1980s. For example, the area of cropland marginally increased by 0.76%, and vegetative areas decreased by 3.19%. The small LULC changes indicated that human activities minimally influenced runoff in the first two decades because LULC changes only contributed to 7.73% of the runoff increase. However, LULC changed considerably with social development and population growth beginning in the 1980s. The vegetative area decreased by 7.81% from the 1980s to the 1990s, whereas the percentages of cropland, barren areas, and urban and built-up areas increased by 2.39%, 5.43%, and 0.11%, respectively. LULC change associated with increased human activities contributed to 54.25% of surface runoff. Furthermore, CCFGP, which was initiated in the late 1990s, mitigated the decreasing trend in vegetation. Although cropland and urban and built-up areas still expanded by 2.40% and 0.82%, respectively, from the 1990s to the 2000s, vegetation increased by 6.00% and barren areas decreased by 9.33%. Therefore, LULC change, exhibited a relatively large influence on the surface runoff change, contributing to 70.67% of surface runoff in the last two decades.

[Figure]

**Figure 9.** Isolated impacts of LULC and climate changes on surface runoff. Positive values indicate that runoff increased due to these factors, whereas negative values indicate that the runoff decreased due to these factors. The summation of the isolated influences is not equal to 100% due to simulation uncertainty (see section 3.6 for details).

Although climate variables were held constant when simulating LULC change, the isolated influence of LULC change on runoff does not exclude the impacts of precipitation variations because the climate (including precipitation) varied in each decade (Table 3). Nonetheless, the above results indicate that LULC change contributed considerably to decreased runoff, as reported in other studies (e.g., Zhang et al., 2011; Zuo et al., 2014; Wang et al., 2014; Wang et al., 2015). Additionally, the results suggest that vegetation restoration due to the CCFGP reduced surface runoff, which is in agreement with the results of other studies (e.g., Li et al., 2009; Qiu et al., 2011; Nunes et al., 2011).

**3.5.2 Impacts of climate change**

Unlike the contribution of LULC change, the influence of climate change decreased in recent decades (Fig. 9). Climate change increased runoff by 26.07 $m^3\,s^{-1}$ from the 1970s to the 1980s, accounting for approximately 87.63% of total runoff in that period. Since the 1980s, surface runoff decreased, and the contributions of climate change to decreased runoff were 55.92% and 42.11% in the 1980s−1990s and 1990s−2000s, respectively. The influence of climate change on runoff is in agreement with the climatic warming and drying trends. The decreasing precipitation trend will potentially lead to less runoff, whereas the increasing temperature will result in increased evaporation.

In summary, LULC and climate changes accounted for 7.73% and 87.63% of the total runoff increase (29.75 $m^3\,s^{-1}$) in the 1970s and 1980s, respectively. The isolated influences of LULC and climate changes on runoff were nearly the same from 1980 to 1999 at 54.25% and 55.92%, respectively, compared to the total runoff decrease. In the last two decades, the percent decrease in total runoff caused by LULC changes (70.67%) was greater than that caused by climate change (42.11%).

Although uncertainties exist in the simulations (see section 3.6 for details), the above results indicated that the contribution of climate variability decreased over the last four decades, while the contribution of LULC change increased. Unlike the results reported by Liang et al. (2015), the findings in this study suggested that runoff fluctuations are influenced less by climate change and more by human activities. The results also indicated that the impact of human activities on runoff has gradually increased in the Jinghe River Basin, which is in agreement with the results of other studies (Zhang et al., 2011; Zuo et al., 2014; Wang et al., 2015).

New reference was also added:

Wang, G., Yang, H., Wang, L., Xu, Z., Xue, B.: Using the SWAT model to assess impacts of land use changes on runoff generation in headwaters, Hydrological Processes, 28, 1032–1042, 2014.

**2.** To explicitly assess the impact of land use changes on runoff generation, the impact of rainfall variation should first be excluded. Relevant discussion should be included on this topic.

**Responses:**

Thank you for the valuable comments. Even the isolated influence of LULC changes on runoff can be simulated by varying LULC while holding climate constant. However, climate (e.g., precipitation) varied in each decade.

According to this comment, we added the following discussion to the revised manuscript:

Although climate variables were held constant when simulating LULC change, the isolated influence of LULC change on runoff does not exclude the impacts of precipitation variations because the climate (including precipitation) varied in each decade (Table 3). Nonetheless, the above results indicate that LULC change contributed considerably to decreased runoff, as reported in other studies (e.g., Zhang et al., 2011; Zuo et al., 2014; Wang et al., 2014; Wang et al., 2015). Additionally, the results suggest that vegetation restoration due to the CCFGP reduced surface runoff, which is in agreement with the results of other studies (e.g., Li et al., 2009; Qiu et al., 2011; Nunes et al., 2011).

**3.** I would encourage authors to rewrite the methodology description section. Give a clear message to the reader what you did and how you did. Some parts in the results analysis and discussion section (e.g. model calibration and validation) are more suitable to be in the methodology section.

**Responses:**

Thank you for this comment. In this study, we simulated runoff change using the SWAT model.

Based on your recommendation, we reorganized section 2 ('Methods and materials'), particularly from sections 2.2 to 2.4 (Table R1), and some related context was also revised.

Table R1 Changes in the structure of section 2

| Structure of section 2 (Methods and materials) in the revised manuscript | Structure of section 2 in the previous manuscript |
|---|---|
| 2 Methods and materials | 2 Methods and materials |
| 2.1 Study area | 2.1 Study area |
| 2.2 Runoff change simulation | 2.2 SWAT model and data collection |
| 2.2.1 SWAT model and data collection | 2.3 Model calibration and validation |
| 2.2.2 Model calibration and validation | 2.4 Runoff change simulation |
| 2.2.3 Simulation scenarios | |

**4.** The values in Table 2 are not acceptable, the authors should re-check the reference (Moriasi et al., 2007).

**Responses:**

We re-checked the reference (Moriasi et al., 2007) and found that the values of the Nash–Sutcliffe coefficient (*Ens*) given in Table 2 in the previous manuscript may cause misleading.

Reference:

Moriasi, D. N., Arnold, J. G., van Liew, M. W., Binger, R. L. Harmel, R. D., and Veith, T.: Model evaluation guidelines for systematic quantification of accuracy in watershed simulations, Trans. Am. Soc. Agr. Biol. Eng., 50, 885–900, 2007.

According to this comment, we revised Table 2 as follows:

**Table 2.** SWAT performance of runoff simulations according to the Nash–Sutcliffe coefficient (Moriasi et al., 2007).

| Simulation performance | Nash–Sutcliffe coefficient (*Ens*) |
|---|---|
| Very good | $0.75 < Ens \leq 1.00$ |
| Good | $0.65 < Ens \leq 0.75$ |
| Satisfactory | $0.50 < Ens \leq 0.65$ |
| Unsatisfactory | $Ens \leq 0.50$ |

**5.** The influences of different land use types (such as area and spatial distribution) are also important to runoff generation. How to assess these effects?

**Responses:**

Yes, the spatial distributions and areas of different land use types can influence runoff generation. We added some discussion regarding how the areas of different land use types affect surface runoff (see our reply to major comment 1). Additionally, previous studies have been conducted to investigate how the spatial distributions of different land use types influence runoff generation (e.g., Wang et al., 2014). Although we can obtain the spatial distributions of different land use types, assessing the influence of spatial changes in land use on runoff is difficult.

Date Issued:
August 30, 2016

Certificate Verification Key:
B618-CD18-89AC-2B8B-C700

[Figure]

This certificate may be verified at www.aje.com/certificate. This document certifies that the manuscript listed above was edited for proper English language, grammar, punctuation, spelling, and overall style by one or more of the highly qualified native English speaking editors at American Journal Experts. Neither the research content nor the authors' intentions were altered in any way during the editing process. Documents receiving this certification should be English-ready for publication; however, the author has the ability to accept or reject our suggestions and changes. To verify the final AJE edited version, please visit our verification page. If you have any questions or concerns about this edited document, please contact American Journal Experts at

Fig. R3 Certificate for English language editing

**Title Page**

Effects of land use/land cover and climate changes on surface runoff in a semi-humid and semi-arid transition zone in Northwest China

Jing Yin [1], Fan He [2], YuJiu Xiong [3, 4, *], GuoYu Qiu [5, *]

[1] Research Center for Sustainable Hydropower Development, China Institute of

Water Resources and Hydropower Research, Beijing 100038, China.

[2] State Key Laboratory of Simulation and Regulation of Water Cycle in River

Basin, China Institute of Water Resources and Hydropower Research, Beijing 100038,

China.

[3] Department of Water Resource and Environments, School of Geography and

Planning, Sun Yat-Sen University, Guangzhou 510275, Guangdong, China.

[4] Key Laboratory of Water Cycle and Water Security in Southern China of

Guangdong High Education Institute, Sun Yat-Sen University, Guangzhou 510275,

Guangdong, China.

[5] Shenzhen Engineering Laboratory for Water Desalinization with Renewable

Energy, School of Environment and Energy, Peking University, Shenzhen 518055,

Guangdong, China.

First author Email address: yinjing@iwhr.com

* Corresponding author: YuJiu Xiong, Email address: xiongyuj@mail.sysu.edu.cn.
Tel./Fax: +86 20 84114575.
Co-corresponding author: GuoYu Qiu, Email address: qiugy@pkusz.edu.cn. Tel./Fax:
+86 755 26033309.

**Abstract**

Water resources, which are considerably affected by land use/land cover (LULC) and climate changes, are a key limiting factor in highly vulnerable ecosystems in arid and semi-arid regions. The impacts of LULC and climate changes on water resources must be assessed in these areas. However, conflicting results regarding the effects of LULC and climate changes on runoff have been reported in relatively large basins such as the Jinghe River Basin (JRB), a typical large catchment ($> 45000 \, \text{km}^2$) located in a semi-humid and arid transition zone on the central Loess Plateau, Northwest China. In this study, we focused on quantifying both the combined and isolated impacts of LULC and climate changes on surface runoff. We hypothesized that under climatic warming and drying conditions, LULC change, which is primarily caused by intensive human activities, such as the conversion of cropland to forest and grassland program (CCFGP), will considerably alter runoff in the JRB. The Soil and Water Assessment Tool (SWAT) was adopted to perform simulations. The simulated results indicated that although runoff increased very little between the 1970s and the 2000s due to the combined effects of LULC and climate changes, LULC and climate changes affected surface runoff differently in each decade, e.g., runoff increased with increased precipitation between the 1970s and the 1980s (precipitation contributed to 88% of the runoff increase). Thereafter, runoff decreased and became increasingly influenced by LULC change, which contributed to 44% of the runoff change between the 1980s and the 1990s and 71% of the runoff change 
[revised manuscript text omitted]
 vegetative areas decreased by 3.19%. The small LULC changes indicated that human activities minimally influenced runoff in the first two decades because LULC changes only contributed to 7.73% of the runoff increase. However, LULC changed considerably with social development and population growth beginning in the 1980s. The vegetative area decreased by 7.81% from the 1980s to the 1990s, whereas the percentages of cropland, barren areas, and urban and built-up areas increased by 2.39%, 5.43%, and

0.11%, respectively. LULC change associated with increased human activities contributed to 54.25% of surface runoff. Furthermore, CCFGP, which was initiated in the late 1990s, mitigated the decreasing trend in vegetation. Although cropland and urban and built-up areas still expanded by 2.40% and 0.82%, respectively, from the

1990s to the 2000s, vegetation increased by 6.00% and barren areas decreased by 9.33%.

Therefore, LULC change, exhibited a relatively large influence on the surface runoff change, contributing to 70.67% of surface runoff in the last two decades.

Although climate variables were held constant when simulating LULC change, the isolated influence of LULC change on runoff does not exclude the impacts of precipitation variations because the climate (including precipitation) varied in each decade (Table 3). Nonetheless, the above results indicate that LULC change contributed considerably to decreased runoff, as reported in other studies (e.g., Zhang et al., 2011;

Zuo et al., 2014; Wang et al., 2014; Wang et al., 2015). Additionally, the results suggest that vegetation restoration due to the CCFGP reduced surface runoff, which is in agreement with the results of other studies (e.g., Li et al., 2009; Qiu et al., 2011; Nunes et al., 2011).

**3.5.2 Impacts of climate change**

Unlike the contribution of LULC change, the influence of climate change decreased in recent decades (Fig. 9). Climate change increased runoff by 26.07 $m^3 s^{-1}$ from the 1970s to the 1980s, accounting for approximately 87.63% of total runoff in that period. Since the 1980s, surface runoff decreased, and the contributions of climate change to decreased runoff were 55.92% and 42.11% in the 1980s−1990s and 1990s−2000s, respectively. The influence of climate change on runoff is in agreement with the climatic warming and drying trends. The decreasing precipitation trend will potentially lead to less runoff, whereas the increasing temperature will result in increased evaporation.

In summary, LULC and climate changes accounted for 7.73% and 87.63% of the total runoff increase (29.75 $m^3 s^{-1}$) in the 1970s and 1980s, respectively. The isolated influences of LULC and climate changes on runoff were nearly the same from 1980 to

1999 at 54.25% and 55.92%, respectively, compared to the total runoff decrease. In the last two decades, the percent decrease in total 
[revised manuscript text omitted]

---

## Referee Comment (RC2) · Anonymous Referee #2 · 4 Nov 2016

General comment: This MS investigates both the combined and isolated impacts of land use/land cover and climate changes on surface runoff in a semi-humid and semi-arid transition zone. I reviewed the discussion paper and a revised version by Xiong on behalf of the authors on Agu. 31, 2016. Most of my concerns are similar to that of the previous reviewer. I found that the authors have addressed most of the previous comments/suggestions and made significant changes to the paper, which has improved its quality considerably. The methods, results, and discussion are now rather well described, and the manuscript may be considered for publication after addressing the following minor concerns. Specific comments: 1. Page 21, line 442. The sentence contains misleading statement because resolution of the soil map is 1:1 000 000

whereas the LULC map has a resolution of 30 m. 2. Citation and reference list. I found last name and publication year of some citations were the same; however, the full citations in the reference section showed different first names, e.g., Wang et al., 2014. In addition, some citations in the reference section were repeated (e.g., page 24 line 521). The authors should proof read the manuscript to avoid such confusions or repetitions. 3. If it possible, can you separate the result and discussion sections. 4. in the line 19, I don't think the Jinghe river is a "large" basin. 5. suggest change the "cropland to forest and grassland program (CCFGP)" to ""Grain for Green" program".

---

## Author Comment (AC2) · 9 Nov 2016

Dear Editor and Reviewers:

Thank you for carefully reading the manuscript and providing constructive suggestions and comments. We appreciate your time and effort in considering the manuscript for publication.

All of the questions/comments will be carefully addressed in the revised manuscript.

The following are point-by-point answers to each question/comment.

Reviewer (s) Comments and our Responses

Reviewer #2

This MS investigates both the combined and isolated impacts of land use/land cover and climate changes on surface runoff in a semi-humid and semi-arid transition zone. I reviewed the discussion paper and a revised version by Xiong on behalf of the authors on Agu. 31, 2016. Most of my concerns are similar to that of the previous reviewer. I found that the authors have addressed most of the previous comments/suggestions and made significant changes to the paper, which has improved its quality considerably. The methods, results, and discussion are now rather well described, and the manuscript may be considered for publication after addressing the following minor concerns.

Specific comments

1. Page 21, line 442. The sentence contains misleading statement because resolution of the soil map is 1:1 000 000 whereas the LULC map has a resolution of 30 m.

Responses: Thank you for the valuable comments. We rewrote the sentence to improve clarity as follows:

In addition, the coarse vegetation information provided by the LULC data in this study can lead to uncertainty in the simulations because vegetation distinction is required in SWAT modelling. Although the LULC data had a relatively high resolution of 30 m, we can only provide a general vegetation categorization, such as forest, due to the data limitation.

2. Citation and reference list. I found last name and publication year of some citations were the same; however, the full citations in the reference section showed different first names, e.g., Wang et al., 2014. In addition, some citations in the reference section were repeated (e.g., page 24 line 521). The authors should proof read the manuscript to avoid such confusions or repetitions.

Responses: Thank you for the valuable comments. We will check the citations carefully,

delete repeate citations, and correct mistakes. For example, the citations of 'Wang et al., 2014' in the text and reference section were revised as follows:

Both climate and land use/land cover (LULC) changes are key factors that can modify flow regimes and water availability (Oki and Kanae, 2006; Piao et al., 2007; Sherwood and Fu, 2014; Wang et al., 2014a).

Nonetheless, the above results indicate that LULC change contributed considerably to decreased runoff, as reported in other studies (e.g., Zhang et al., 2011; Zuo et al., 2014; Wang et al., 2014b; Wang et al., 2016).

Wang, R., Kalin, L., Kuang, W., and Tian, H.: Individual and combined effects of land use/cover and climate change on Wolf Bay watershed streamflow in southern Alabama, Hydrological Processes, 28, 5530–5546, 2014a.

Wang, G., Yang, H., Wang, L., Xu, Z., Xue, B.: Using the SWAT model to assess impacts of land use changes on runoff generation in headwaters, Hydrological Processes, 28, 1032–1042, 2014b.

3. If possible, can you separate the result and discussion sections.

Responses: Thank you for the valuable comments. Combined with comments from another reviewer and the Editor, we will separate section 3 ('Results and Discussion') into two sections as follows, including section 3 ('Results') and section 4 ('Discussion'), and related context will also be revised.

3 Results

3.1 Climate change

3.2 LULC change

3.3 Performance of the SWAT model

3.4 Simulated surface runoff

4 Discussion

4.1 Impacts of LULC and climate changes on surface runoff

4.1.1 Isolated impacts of LULC change

4.1.2 Isolated impacts of climate change

4.2 Uncertainty in SWAT model simulations

4. In the line 19, I don't think the Jinghe river is a "large" basin.

Responses: We agree that the term "large" is difficult to determine quantitatively. According to this comment, we revised the sentence as follows:

However, conflicting results regarding the effects of LULC and climate changes on runoff have been reported in relatively large basins such as the Jinghe River Basin (JRB), a typical catchment (> 45000 km2) located in a semi-humid and arid transition zone on the central Loess Plateau, Northwest China.

5. suggest change the "cropland to forest and grassland program (CCFGP)" to "Grain for Green" program.

Responses: Thank you for this comment. We replaced "cropland to forest and grassland program (CCFGP)" with "Grain for Green Program" in the entire manuscript.

---

## Author Response (AR1)

**Reply to Editor and Reviewers**

**Manuscript title**: Effect of land use/land cover and climate changes on surface runoff in a semi-humid and semi-arid transition zone in Northwest China

Dear Editor and Reviewers:

Thank you for carefully reading the manuscript and for providing constructive suggestions and comments. We appreciate your time and effort in considering the manuscript for publication.

All of the questions/comments have been carefully addressed in the revised manuscript. In this revision, the newly added content is given in blue, and the revised content is given in red.

The following are point-by-point answers to each question/comment.

**Editor & Reviewer Comments and our Responses**

**Editor Comments:**

**1.** separate Results and Discussion.

**Responses:**

Thank you for your valuable comments.

Considering the comments from all of the reviewers, we separated section 3 ('Results and Discussion') into two sections (Table R1): section 3 ('Results') and section 4 ('Discussion'), and revised the related content. Please see the revised manuscript for details.

Table R1 Structure changes due to separation of section 3 ('Results and Discussion').

| Revised structure in the manuscript | Structure of section 3 in the previous manuscript |
|---|---|
| 3 Results | 3 Results and Discussion |
| 3.1 Climate change | 3.1 Climate change |
| 3.2 LULC change | 3.2 LULC change |
| 3.3 Performance of the SWAT model | 3.3 Performance of the SWAT model |
| 3.4 Simulated surface runoff | 3.4 Impacts of LULC and climate changes on surface runoff |
| 4 Discussion | 3.5 Uncertainty in model simulations |
| 4.1 Impacts of LULC and climate changes on surface runoff | |
| 4.1.1 Isolated impacts of LULC change | |
| 4.1.2 Isolated impacts of climate change | |
| 4.2 Uncertainty in SWAT model simulations | |

**2.** The authors have to make insightful analyses and discussion on the mechanism behind so that highlight the scientific merit of the paper, otherwise the study is trivial.

**Responses:**

Thank you for your valuable comments.

As shown in Table R1, we reorganized our manuscript and discussed: 1) 'the effects of LULC and climate changes on surface runoff' including the 'Isolated impacts of LULC change' and the 'Isolated impacts of climate change', and 2) the simulation uncertainty in the context of SWAT modelling due to parameterizations. The added discussion provides potential explanations for the conflicting results regarding the effects of LULC and climate changes on runoff in relatively large basins.

Please see our revised manuscript for details as well as our response to the reviewers' comments.

**Reviewer: 1**

This manuscript investigates the relationships between land use and climate changes and corresponding hydrological responses in northwest China. The paper reveals some interesting findings. The manuscript can be considered for publication, if the following comments can be addressed properly.

**Major comments**

1. The effects of land use change and climate change should be discussed separately.

    **Responses:**

    Thank you for the valuable comments.

    Combined with comments from another reviewer and the Editor, we separated section 3 ('Results and Discussion') into two sections: section 4.1.1 ('Isolated impacts of LULC change') and section 4.1.2 ('Isolated impacts of climate change'), and discussed 'the effects of LULC and climate changes on surface runoff') separately (Table R1). The new sections discussing the effects of land use and climate changes are as follows:

    Table R1 Structure changes due to separation of section 3 ('Results and Discussion')

[revised manuscript text omitted]

**2.** To explicitly assess the impact of land use changes on runoff generation, the impact of rainfall variation should first be excluded. Relevant discussion should be included on this topic.

**Responses:**

Thank you for the valuable comments. Even the isolated influence of LULC changes on runoff can be simulated by varying LULC while holding climate constant. However, the climate (e.g., precipitation) varied in each decade.

To address this comment, we added the following discussion in the revised manuscript:

Although climate variables were held constant when simulating LULC changes, the isolated

influences of LULC changes on runoff did not exclude the impacts of precipitation variations because the climate (including precipitation) varied in each decade (Table 3). Nonetheless, the above results indicate that LULC changes contributed considerably to decreased runoff, as reported in previous studies (e.g., Zhang et al., 2011; Zuo et al., 2014; Wang et al., 2014b; Wang et al., 2016). Additionally, the results suggest that vegetation restoration due to the GGP reduced surface runoff, which agrees with the results of other studies (e.g., Li et al., 2009; Nunes et al., 2011).

**3.** I would encourage authors to rewrite the methodology description section. Give a clear message to the reader what you did and how you did. Some parts in the results analysis and discussion section (e.g. model calibration and validation) are more suitable to be in the methodology section.

**Responses:**

Thank you for this comment. In this study, we simulated runoff changes using the SWAT model.

Based on your recommendation, we reorganized section 2 ('Methods and materials'), particularly sections 2.2 to 2.4 (Table R2), and revised some of the related content.

Table R2 Changes in the structure of section 2

| Structure of section 2 (Methods and materials) in the revised manuscript | Structure of section 2 in the previous manuscript |
|---|---|
| 2 Methods and materials | 2 Methods and materials |
| 2.1 Study area | 2.1 Study area |
| 2.2 Runoff change simulation | 2.2 SWAT model and data collection |
| 2.2.1 SWAT model and data collection | 2.3 Model calibration and validation |
| 2.2.2 Model calibration and validation | 2.4 Runoff change simulation |
| 2.2.3 Simulation scenarios | |

**4.** The values in Table 2 are not acceptable, the authors should re-check the reference (Moriasi et al., 2007).

**Responses:**

We re-checked the reference (Moriasi et al., 2007) and found that the values of the Nash–Sutcliffe coefficient (*Ens*) given in Table 2 in the previous manuscript were misleading.

Reference:

Moriasi, D. N., Arnold, J. G., van Liew, M. W., Binger, R. L. Harmel, R. D., and Veith, T.: Model evaluation guidelines for systematic quantification of accuracy in watershed simulations, Trans. Am. Soc. Agr. Biol. Eng., 50, 885–900, 2007.

According to this comment, we revised Table 2 as follows:

**Table 2.** SWAT performance of runoff simulations according to the Nash–Sutcliffe coefficient (Moriasi et al., 2007).

| Simulation performance | Nash–Sutcliffe coefficient ($Ens$) |
|---|---|
| Very good | $0.75 < Ens \leq 1.00$ |
| Good | $0.65 < Ens \leq 0.75$ |
| Satisfactory | $0.50 < Ens \leq 0.65$ |
| Unsatisfactory | $Ens \leq 0.50$ |

**5.** The influences of different land use types (such as area and spatial distribution) are also important to runoff generation. How to assess these effects?

**Responses:**

Yes, the spatial distributions and areas of different land use types can influence runoff generation. Some studies (e.g., Wang et al., 2014) have been conducted to investigate how the spatial distributions of different land use types influence runoff generation. Although we can obtain the spatial distributions of different land use types, assessing the influences of spatial changes in land use on runoff is difficult.

**Date Issued:**
August 30, 2016

[Figure]

**Certificate Verification Key:**
B618-CD18-89AC-2B8B-C700

This certificate may be verified at www.aje.com/certificate. This document certifies that the manuscript listed above was edited for proper English language, grammar, punctuation, spelling, and overall style by one or more of the highly qualified native English speaking editors at American Journal Experts. Neither the research content nor the authors' intentions were altered in any way during the editing process. Documents receiving this certification should be English-ready for publication; however, the author has the ability to accept or reject our suggestions and changes. To verify the final AJE edited version, please visit our verification page. If you have any questions or concerns about this edited document, please contact American Journal Experts at support@aje.com.

Fig. R3 Certificate for English language editing

**Reviewer: 2**

This MS investigates both the combined and isolated impacts of land use/land cover and climate changes on surface runoff in a semi-humid and semi-arid transition zone. I reviewed the discussion paper and a revised version by Xiong on behalf of the authors on Agu. 31, 2016. Most of my concerns are similar to that of the previous reviewer. I found that the authors have addressed most of the previous comments/suggestions and made significant changes to the paper, which has improved its quality considerably. The methods, results, and discussion are now rather well described, and the manuscript may be considered for publication after addressing the following minor concerns.

**Specific comments**

1. Page 21, line 442. The sentence contains misleading statement because resolution of the soil map is 1:1 000 000 whereas the LULC map has a resolution of 30 m.

**Responses:**

Thank you for your valuable comments. We rewrote this sentence as follows:

In addition, the coarse vegetation information provided by the LULC data in this study can lead to uncertainty in the simulations because vegetation distinction is required in SWAT modelling. Although the LULC data had a relatively high resolution of 30 m, we can only provide a general vegetation categorization, such as forest, due to the data limitation.

2. Citation and reference list. I found last name and publication year of some citations were the same; however, the full citations in the reference section showed different first names, e.g., Wang et al., 2014. In addition, some citations in the reference section were repeated (e.g., page 24 line 521). The authors should proof read the manuscript to avoid such confusions or repetitions.

**Responses:**

Thank you for the valuable comments. We checked the citations carefully, deleted repeated citations, and corrected mistakes. For example, the citations of 'Wang et al., 2014' in the text and reference section were revised as follows:

Both climate and land use/land cover (LULC) changes are key factors that can modify flow regimes and water availability (Oki and Kanae, 2006; Piao et al., 2007; Sherwood and Fu, 2014; Wang et al., 2014a).

Nonetheless, the above results indicate that LULC change contributed considerably to decreased runoff, as reported in other studies (e.g., Zhang et al., 2011; Zuo et al., 2014; Wang et al., 2014b; Wang et al., 2016).

Wang, R., Kalin, L., Kuang, W., and Tian, H.: Individual and combined effects of land use/cover and climate change on Wolf Bay watershed streamflow in southern Alabama, Hydrological Processes, 28, 5530–5546, 2014a.

Wang, G., Yang, H., Wang, L., Xu, Z., Xue, B.: Using the SWAT model to assess impacts of land use changes on runoff generation in headwaters, Hydrological Processes, 28, 1032–1042, 2014b.

**3.** If possible, can you separate the result and discussion sections.

**Responses:**

Thank you for the valuable comments.

Combined with comments from another reviewer and the Editor, we separated section 3 ('Results and Discussion') into two sections (Table R1): section 3 ('Results') and section 4 ('Discussion'), and revised the related content.

Table R1 Structure changes due to separation of section 3 ('Results and Discussion').

| Revised structure in the manuscript | Structure of section 3 in the previous manuscript |
| --- | --- |
| 3 Results | 3 Results and Discussion |
| 3.1 Climate change | 3.1 Climate change |
| 3.2 LULC change | 3.2 LULC change |
| 3.3 Performance of the SWAT model | 3.3 Performance of the SWAT model |
| 3.4 Simulated surface runoff | 3.4 Impacts of LULC and climate changes on surface runoff |
| 4 Discussion | 3.5 Uncertainty in model simulations |
| 4.1 Impacts of LULC and climate changes on surface runoff | |
| 4.1.1 Isolated impacts of LULC change | |
| 4.1.2 Isolated impacts of climate change | |
| 4.2 Uncertainty in SWAT model simulations | |

**4.** In the line 19, I don't think the Jinghe river is a "large" basin.

**Responses:**

We agree that the term "large" is not very quantitative.

According to this comment, we revised this sentence as follows:

However, conflicting results regarding the effects of LULC and climate changes on runoff have been reported in relatively large basins, such as the Jinghe River Basin (JRB), which is a typical catchment ($> 45000\,\mathrm{km}^2$) located in a semi-humid and arid transition zone on the central Loess Plateau, Northwest China.

**5.** suggest change the "cropland to forest and grassland program (CCFGP)" to "Grain for Green"

program.

**Responses:**

Thank you for this comment. We replaced "cropland to forest and grassland program (CCFGP)" with "Grain for Green Program" throughout the manuscript. Please see our revised manuscript for details.

The end